# Evaluating Culturalization Strategies for Sustainable Tourism Development in Uzbekistan

**Ji Young Jeong [1], Mamurbek Karimov [1], Yuldoshboy Sobirov [1], Olimjon Saidmamatov [2],* and Peter Marty [3],***

1 Department of International Trade, Jeonbuk National University, Jeonju-si 54896, Republic of Korea; j3021@jbnu.ac.kr (J.Y.J.); mamurbek@jbnu.ac.kr (M.K.); syuldoshboy@jbnu.ac.kr (Y.S.)
2 Faculty of Economics, Urgench State University, Urgench 220100, Uzbekistan
3 Institute of Natural Resource Sciences, Zurich University of Applied Sciences (ZHAW), 8820 Wädenswil, Switzerland
* Correspondence: saidolimjon@gmail.com (O.S.); marp@zhaw.ch (P.M.)

**Abstract:** Tourism is one of the fastest-growing and most visible sectors of the global economy, contributing significantly to boosting a nation's economy. Currently, tourism-led economic growth considering sustainable approaches is becoming increasingly important in public policy. Recent urban and regional studies have begun to focus on the sustainability of tourism from social, economic, environmental, and cultural perspectives. The research aims to identify the most important issues and barriers to sustainable tourism development in Uzbekistan and proposes the most effective culturalization strategies to overcome those barriers. This paper involves the application of Global RPM (Globalization, Rationality, Professionalism, and Morality) analysis in combination with SANEL HERMES (Sightseeing, Admission paying, Night touring, Experiencing, Learning, Healing, Enjoying, Rest and Relaxing, Memento shopping, Eating and Drinking, and Staying) model based on data from a variety of sources, including literature review, participation interview, tourist survey, and expert questionnaire to identify and classify influencing factors that show existing barriers to sustainable tourism in Uzbekistan. Finally, a set of culturalization strategies is proposed, evaluated, and ranked by experts using a Quantitative Strategic Planning Matrix. The results of the survey have shown that tourism activities are currently not sustainable in a satisfactory manner. However, if the policy-makers consider the relevant strategies and take this study into account, as well as its findings, then Uzbekistan can be made more attractive in terms of sustainable tourism.

**Keywords:** sustainable tourism; culturalization strategies; Global RPM; SANEL HERMES; QSPM

## 1. Introduction

In many parts of the world, tourism is a major economic activity [1–5], creating jobs in large numbers [6] and attracting investments and foreign capital [7]. As well as promoting and developing peace, prosperity, and national and international relationships, tourism can contribute to the realization of sustainable development goals [8]. The tourism industry has emerged as a key force for sustainable socioeconomic development globally [9,10]. Essentially, sustainable tourism refers to tourism that does not harm nature and the local community and also contributes to the country's environment, society, and economy in a positive way [11]. There is a broad range of activities associated with tourism, including accommodation, transportation, entertainment, leisure, nourishment, and shopping, being related to travel aims for business, recreation, medicine, and visiting friends or relations [10,12]. Currently, most agree that tourism's growth needs to be sustainable, although the question of how to achieve this is a subject of debate [13].

The importance of sustainability is being increasingly recognized by the world, so all industries are encouraged to come up with their own sustainable solutions. Sustainability focuses its "attention on a set of ethical values and principles, which guides action in a

responsible and harmonious way, incorporating the environmental and societal consequences of actions, as well as economic goals" [14]. Several fields are currently using the term 'sustainable', such as the development of communities, agriculture, architecture, and tourism [15]. Many industries have therefore begun to prioritize sustainability in their strategic planning [16]. As an example, tourism, which is the fastest-growing industry in the world, has been taking steps to become a sustainable industry by developing responsible methods [17]. Additionally, tourism, an essential industry for both social and economic development, can positively affect production, gross revenue, and employment. Alternatively, it can negatively impact the environment [18]. It is important to note that in the event of poorly planned and managed tourism, there may be permanent damage to a tourist destination's economic, social, physical, and cultural environments [19]. In this regard, sustainable tourism must be emphasized as an important issue. A number of issues, such as the failure to preserve natural, historical, social, and cultural resources, the loss of social values, and the disruption of the ecological balance caused by global warming, make sustainable tourism an essential part of any tourism strategy [20].

As a goal of sustainable tourism, culturalization strategies have an important role in sustainable development, where sustainable and long-term achievements are more important than short-term achievements in conserving natural beauty and cultural values [21,22]. Culturalization strategies are not only designed to generate income, but they also protect and preserve local assets, history, local traditions, and customs by providing economic support for maintaining those assets and traditions. In recent decades, indicators, data, and statistics on the cultural sector have demonstrated that culture can play an important role in socioeconomic, environmental, and cultural development. It is common in the tourism industry for the culture of the destination to be considered a tourism investment, as well as any changes, such as the opening of a region to tourists, must also be taken into account when planning any tourism project [23] and cultural resources are seen as one of the most important aspects of any tourist attraction [24,25]. As a powerful socioeconomic resource, culture can be incorporated into sustainable development strategies and policies, promoting an inclusive and humanistic development perspective. There is no doubt that culture plays a significant role in sustainable development, but as part of that consideration, it is imperative to give due attention to both the process and the outcome. For this reason, the choice of culturalization strategies is rational for the subject of sustainable tourism because they provide a framework for managing tourism in a way that is respectful of cultural heritage, promotes intercultural understanding, supports the conservation and preservation of cultural sites, assets, and products, as well as contributing to the economic development of local communities. Culturalization strategies for sustainable tourism can help to address some of the negative impacts of tourism on cultural heritage sites and communities. These impacts can include over-tourism, degradation of cultural sites, and erosion of traditional cultural practices and values. Culturalization strategies can provide guidance on how to manage tourism in a way that minimizes these impacts while maximizing the positive benefits of tourism to be a more peaceful, tolerant, and sustainable world that can be preserved for future generations.

With its ancient and cultural heritage, as well as its natural attraction, Uzbekistan is able to expand its tourism sector. British publication The Telegraph named Uzbekistan as one of the top 10 beautiful and amazing travel destinations for 2019 [26]. Furthermore, The Telegraph points out that Uzbekistan, home to the Silk Road, offers many reasons to visit, with Bukhara, Khiva, Shakhrisabz, and Samarkand as the main UNESCO-recognized tourist hubs. In fact, Uzbekistan is one of the world's 30 safest countries, according to the Global Terrorism Index-2019 [27]. It is estimated that Uzbekistan is home to almost 7400 objects of cultural heritage [28]. Specifically, Uzbekistan is attracting more international tourists, with there being 6.7 million international tourists in 2019 as compared to 1.9 million people in 2014 [29]. Based on the above-mentioned information, Uzbekistan's tourism sector is growing rapidly and has significant economic importance.

In particular, despite the significance of analyses to explore and evaluate future strategies for tourism sustainability in the previous literature [14,17,19,23,30–32], no previous research has been conducted to explore sustainable tourism development in Uzbekistan, and no efforts have been made to identify strategies through Global RPM and QSPM analysis for contribution to tourism sustainability. This represents a significant research gap in the sustainable tourism literature of Uzbekistan. Therefore, our primary objective is to identify the most important issues and barriers to sustainable tourism development in Uzbekistan and propose the most effective strategies to overcome those barriers. The second objective is to develop a methodology that presents a strategic approach to the contribution of tourism sustainability that can be applied to different tourist destinations. In order to accomplish these objectives, the following research questions needed to be answered:

Is the tourism industry currently engaged in sustainable activities?
What kind of obstacles does the country's tourism industry experience when it comes to achieving sustainable development?
Which culturalization strategies are positioned to be most effective for sustainable tourism development in Uzbekistan?
In order to successfully boost long-term sustainability in the tourism sector, what kinds of government measures should be implemented first?

Accordingly, to obtain answers to the above questions, the Global RPM Analysis includes analyzing an acronym of globalization, rationality, professionalism, and morality as factors that can affect a business or organization, and SANEL HERMES that involves identifying an acronym of Sightseeing, Admission paying, Night touring, Experiencing, Learning, Healing, Enjoying, Rest and relaxing, Memento shopping, Eating and drinking and Staying for a tourism destination, the hybrid method will be used in this study. Specifically, both factors of Global RPM affecting the tourism industry in Uzbekistan will be ranked by tourists and prioritized based on the importance of the results to evaluate the suitability of culturalization strategies. Global RPM analysis provides a framework for decision-making that takes into account multiple factors, which encourages a holistic approach to business evaluation rather than focusing on just one or two factors. This can help businesses to consider all aspects of their operations, make informed decisions and ensure that they are addressing all potential risks and opportunities. Furthermore, a Quantitative Strategic Planning Matrix (QSPM) will be used to formulate the strategy proposals for sustainable tourism in Uzbekistan, along with a region-specific vision statement. Although the study is limited to Uzbekistan, the research presented in the developed paper has significant relevance to the international scientific field, as it provides new knowledge and insights into a relevant research problem, has the novelty of methodology and potential practical applications, and can inform policy and practice in various global contexts. Furthermore, in the paper developed, authors have extensively investigated a novel approach to solve a significant problem in the field of sustainable tourism. The research methodology used is systematic and comprehensive, with appropriate controls and basic data to support the findings, which have significant implications for the field with the potential to provide new insights into sustainable tourism. The findings are supported by relevant literature and are likely to spark further research and discussion in the field. It could also prove useful for similar studies in other countries to utilize the study's methodology and framework to analyze the effectiveness of strategies to promote sustainable tourism. The overall approach, however, can be adapted to suit other countries' purposes despite the specific results of this study not being applicable.

There are six sections in this research. The introduction to the paper is dealt with in Section 1. An overview of the literature is presented in Section 2. The methods and database used in the study are described in Section 3. The main results of the study are detailed in Section 4, along with strategies to overcome the barriers. The results of the research are discussed in Section 5. As a final section, the study's conclusion is provided, along with its contributions and limitations.

## 2. Literature Review

Numerous authors have discussed sustainable tourism as a topic [30,33–36]. Nevertheless, relatively few studies have been conducted on culturalization strategies for the development of sustainable tourism. As a first step, it would be beneficial to briefly review the literature regarding sustainable tourism development strategies in order to better understand the purpose of this study.

The development strategies of sustainable tourism have been studied in different regions. A sustainability assessment of the Egyptian tourism planning mechanism was carried out by [37]. He showed that programs for the development of sustainable tourism are lacking in the planning system for tourism in Egypt, and the tourism sector needs to cooperate more with the government to succeed. Marketing strategies for sustainable tourism were formulated for Barcelona [38]. The authors highlighted the need to reduce carbon emissions from transportation, normalize the behavior of tourists, minimize social and environmental consequences, and compensate for implications for the negative that tourism may have on the environment, human health, and culture. Ref. [39] provided another example of how the Bolivian government adopts sustainable tourism strategies to improve the state's economy. As she noted, sustainable tourism development must be aligned with the community's self-actualization and contemplated in advance. Geographic Information System (GIS) and The Analytic Network Process (ANP) were used for sustainable tourism development in Malaysia's Cameron Highlands [40]. By integrating GIS and ANP, the researchers have shown that the analytical tools can provide useful assistance in spatial planning in terms of sustainable tourism. An empirical study [41] examined the possibility of enhancing sustainable tourism by combining the Fuzzy Set Theory with the Analytic Hierarchy Process (AHP) and demonstrating its application to Taiwan's Green Island. According to the authors cited above, tourism development is potentially beneficial for the economy, but it can also negatively affect the environment, which is why a well-crafted plan for tourism development is essential in order to maintain the balance between the two. Additionally, they highlighted that more attention should be paid to environmental and ecological protection by the tourism authorities.

Though there have been many studies contributing to sustainable tourism development [42–46], neither the Global RPM nor SANEL HERMES models have yet been published or used with QSPM. However, there is some literature in which the authors have applied alternative methods to the models to analyze sustainable tourism and formulate strategies based on QSPM or a strategic management tool. Several scholars have used QSPM together in combination with SWOT and PESTEL analysis in order to compare different strategies and determine which of them is the most advantageous [47,48]. In the case of the sustainable ecotourism development of Rameswaram city in India, using SWOT and QSPM approach, ref. [49] presented that visitors' satisfaction level was low about services, lodging, unique food habits, and pollution in the city. Ref. [50] used PESTEL analysis and QSPM to analyze local people's and farmers' perspectives on rural tourism. They showed that the level of materiality intertwined with the PESTEL sizes of rural tourism, and each dimension was important for the perception of rural tourism both with each other and alone. Ref. [51] conducted research regarding the strategies to develop the tourism sector on Changbai Mountain. To determine the optimum strategy, the combined SWOT-AHP model with four criteria and 28 sub-criteria was used, with offensive methods being prioritized over defensive methods. According to their results, managers and planners must give special consideration to the tourism potential of this region to strengthen the tourism industry and the local economy and create jobs. Despite the methods that can be alternatives to Global RPM analysis and SANEL HERMES of tourism model, they differ from others for several reasons, such as spheres, features, opportunities, and purposes of use. This paper will contribute literature on strategic planning techniques as well as provide brief definitions of the models in the following chapter.

Compared to the literature on sustainable strategies, which has grown steadily and contains commonly known discourses, the literature on culturalization strategies requires

further discussion. Thus, one of the main purposes of this research is to describe and offer culture-related strategies for sustainable tourism. The term culturalization can be defined to identify culturalization strategies. Refs. [52,53] defines culturalization as "the process by which a specific community transforms its economic activities and empowers its members by developing products and services based on its particular cultural and geographical environment". In addition, ref. [54] described that "a culturalization strategy is the tool that local and regional policymakers have to boost their cities' or territories' comparative advantage and competitiveness through their existing or potential cultural endowment". He mentioned that culture could contribute to sustainable development through some suggested measures that aim to integrate traditional environmental practices with high-tech advances while also supporting culture-based urban revitalization, sustainable cultural tourism, cultural and creative industries, and cultural institutions.

Culturalization strategies are referred to in the literature [55,56] as the process through which sustainable development is promoted through the integration of culture. Within previous pioneering research of the strategies, ref. [53] conducted an assessment of the commercialization and culturalization of Mayan traditions and culture throughout the world. They presented strategies for developing cultural tourism as an effective source of income for Guatemala, identifying sustainable tourism as well as analyzing rural community participation for its integration and development. Ref. [57] applied culturalization strategies for community tourism in the Ecuadorian Amazon, and showed a classification by type of the indigenous people who live and work there through Community Tourism. Besides using the tourism industry, ref. [58] explain what culturalization is, how it influences communities and economic performance with strong cultural elements, and show how these strategies have been applied to the coffee industry in Guatemala. According to the results of a comparative study in Bucharest and Paris [59], a comprehensive analysis of cultural tourism categories and subcategories is essential to identify ways to promote and capitalize on cultural tourism and make it more sustainable, particularly in Bucharest.

It is important to note that foreign tourists were partially restricted from entering Uzbekistan for more than a decade, but now the government has begun to welcome tourists and facilitate tourism development by allowing foreigners to access Uzbekistan's rich cultural and historical heritage [28]. Currently, the tourism sector has a high potential for growth in the view of the government of Uzbekistan. Uzbekistan's tourism industry is undervalued in comparison to its potential and being home to ancient archaeological and cultural sites and has not received the rights and consideration it deserves [60]. It is apparent that there is a significant gap between demand and service delivery. There are a few studies for Uzbekistan to conduct detailed analysis in order to come up with an optimal solution to the issue of tourism in the country. Nevertheless, there have been relatively few studies that have examined Uzbekistan's sustainable tourism in depth and evaluated its economic, social, cultural, and environmental impacts. Therefore, this paper suggests a comprehensive analysis is best for sustainable tourism in Uzbekistan.

Sustainable tourism and culturalization strategies are discussed in the literature section, which includes case studies from various regions by applying a variety of approaches and techniques, such as benchmarking, AHP, GIS, ANP, fuzzy logic, SWOT, PESTEL, and QSPM. In general, it can be said that these studies categorize sustainable tourism and culturalization strategies as environmental, social, and economic. A number of authors have attempted to evaluate the existing tourism planning of the regions and put forward suggestions for sustainable tourism that can be implemented in the regions in the future. Furthermore, there have been a number of authors that have highlighted the crucial role of experts and stakeholders, as well as the significance of a holistic perspective or a systematic approach to the development of sustainable tourism. Based on the purpose of this study, an integrated approach to the development of sustainable tourism in Uzbekistan is demonstrated by using the application of Global RPM analysis and the SANEL HERMES model, and the QSPM with regard to the vision statement designed for the region and sustainable tourism goals.

## 3. Methodology

This research is conducted in Uzbekistan, which is located in Central Asia and bordered by five landlocked countries: Turkmenistan to the southwest, Kyrgyzstan to the northeast, Tajikistan to the southeast, Kazakhstan to the north, and Afghanistan to the south. Compared to other Central Asian countries, Uzbekistan has a particularly rich natural tourism resource base, and there are still plenty of tourism opportunities that can still be developed, such as cultural tourism, culinary tourism, wildlife tourism, rural tourism, dark tourism, and adventure tourism. This study aims to identify the most important issues and barriers and provide assessment and optimization for culturalization strategies designed to enhance sustainable tourism development in Uzbekistan. In this regard, the data presented in the study can be practical and relevant to a wide range of the audience, including policymakers and government officials, to assist in planning policies, regulations, and programs for addressing environmental, social, and economic challenges of sustainable tourism development; academics and researchers to develop new research questions and methodologies; tourism industry stakeholders such as hotels, restaurants, tour operators, and transportation companies to understand the preferences and behaviors of tourists related to sustainable tourism practices and to adjust their services accordingly; business and industry leaders to develop sustainable business models, products, and services; and the general public to make more informed decisions about their consumption patterns, lifestyle choices, and civic engagement, and to contribute to protecting natural and cultural resources, and a more sustainable future for all. Overall, the presented study of sustainable development can be applicable to anyone involved in the tourism industry, from businesses and government agencies to tourists themselves, as it provides valuable insights into the attitudes and preferences of sustainable travelers.

### 3.1. Methods

This study performs a hybrid analysis of the Global RPM and SANEL HERMES, and QSPM and QSPM for developing sustainable tourism in the region while being applicable to situations of strategic planning. Therefore, the models together enable policymakers to gain an in-depth understanding of globalization, rationality, professionalism, and morality of the tourism sector, which is important because it facilitates strategic planning, enables better risk management, supports innovation, enhances communication and collaboration, and helps to build sustainable development. The tourism sector can gain a deeper understanding of the positives and negatives of the environment using both frameworks since they assess the environment from different perspectives.

3.1.1. Global RPM Analysis

The Global RPM analysis technique was found by Professor Jeong Ji Young at Jeonbuk National University in 2018 [61]. Global RPM is an acronym for Globalization, Rationality, Professionalism, and Morality which is a framework involved in a project/business, such as a product line or division, an industry, or another entity, which enables a group/individual to organize and evaluate the important factors linked to success and failure in the business world using global, rational, professional, and moral assessment to be more competitive in today's market and look at global and traditional strategies from a new perspective.

Global RPM analysis allows businesses to understand and evaluate their potential to become an international brand and globalize their business. It also helps businesses to establish and benefit from a rational decision-making process that seeks to maximize utility. By considering professionalism factors, businesses can assess their level of competence and adherence to a set of standards and guidelines. Finally, by considering morality factors, businesses can ensure they are behaving in a socially responsible and ethical manner towards the environment and people. Overall, Global RPM analysis helps businesses to identify advantages, disadvantages, limitations, and influences, which allows them to reduce the chances of failure and become more competitive in the global market. It provides a well-rounded view of the many factors that could affect a business and helps

businesses to make informed decisions based on a broad range of considerations. Therefore, understanding every aspect of a business instead of focusing only on rational factors can reduce the chances of failure in the future. The dimensions of Global RPM can be defined following:

- ➤ Globalization factors—refer to the methods by which a business/a company can succeed in different markets and places. General suggesting keywords of globalization can include Global index (peace, war, environment, disease, language, study abroad), international infrastructure, the internet, international tourism, and so on.
- ➤ Rationality factors—are described to be an economic decision-making process that seeks to maximize utility. In this dimension, other methods or theories can be used to understand the rationality concepts of a business. For instance, empirical analyses and theoretical analyses, such as SWOT or PESTLE, can be applied to assess the rationality of a business as a part of Global RPM. General suggesting keywords of rationality can include the law of nature, economic theory, SWOT, empirical model, demand–supply analysis, consistency (as an example, if a > b, b > c, and then a > c), efficiency, and so on.
- ➤ Professionalism factors—are competence, degree, or skill expected of a professional as well as adhering to a set of standards and guidelines or collection of characteristics that distinguish acceptable practices in a specific field. General suggesting keywords of professionalism can include certificates, experience, patents, awards, records, degrees, proficiency, professional uniqueness, and so on.
- ➤ Morality factors—are a set of standards that provide principles for how companies, businesses, individuals, and groups are to behave and deal with the environment and people or institutions. General suggesting keywords of morality can include Ethics, Manner, Environmental, Social, and Governance (ESG), standards, principles, characteristics, adoption, and so on.

By incorporating the four dimensions of globalization, rationality, professionalism, and morality, businesses can become more competitive in the global market and promote responsible and sustainable business practices. The methodology involves a step-by-step approach to analyzing a business or project in terms of its potential impact. The first step in the Global RPM methodology is to identify the scope and goals of the analysis and areas of improvement. This involves determining the boundaries of the business or project being analyzed and defining the goals and objectives of the analysis. Once the goals are identified, the second step is to evaluate the factors that could influence whether or not the business or project can succeed. These factors are analyzed using the four dimensions of Global RPM: globalization, rationality, professionalism, and morality. The third step is to develop strategies for addressing the factors identified in the analysis. This involves identifying opportunities for growth and improvement, as well as potential threats and challenges. The final step is to implement the strategies developed in the previous step. As strategies are implemented, progress is monitored, and adjustments can be made as necessary.

Global RPM analysis differs from other planning tools in that the four dimensions of Global RPM can be used together or individually to evaluate an aspect of a business. There is no single most important dimension, as any dimension can be used on any segment of the economy to find profitability and attractiveness. Importantly, Global RPM analyses and monitors the macro–micro environmental factors that have an impact on a company. It is beneficial to have a well-rounded view of the many factors that could affect businesses. Global RPM analysis can facilitate strategic planning by helping businesses to develop short-term and long-term plans that are based on a thorough evaluation of different factors. It is important for tourism businesses to evaluate the globalization dimension in order to assess the potential for international growth and to develop strategies for attracting visitors from around the world. However, tourism authorities must also consider the environmental impact of increased tourism and develop strategies for minimizing their carbon footprint and protecting natural resources. By using the analysis to evaluate influencing factors, tourism businesses can develop sustainable tourism practices that balance economic growth

with environmental and social responsibility. This can help to ensure the long-term success of tourism businesses and destinations to be more proactive in their decision-making and better positioned while also contributing to the overall sustainability of the tourism industry. For this reason, Global RPM analysis is performed for the study in order to identify advantages, disadvantages, limitations, and influences.

3.1.2. SANEL HERMES of Tourism Model

SANEL HERMES model, which studies the key factors influencing tourism destinations, is an acronym for Sightseeing (S), Admission paying (A), Night touring (N), Experiencing (E), Learning (L), Healing (H), Enjoying (E), Rest and Relaxing (R), Memento shopping (M), Eating and Drinking (E), and Staying (S), respectively (neither commercial nor religious meanings are intended). The tourism model was introduced by Professor Jeong Ji Young at Jeonbuk National University in 2021 [61].

The components of this model are important to consider when tourism managers are developing a destination, so they can make sure that all the components are suitable to the quality and demand of visitors. Furthermore, tourists can consider these components as primary requirements when making travel plans for various tourism-related activities.

A tourism item is anything that visitors require while they are away from home, including facilities, activities, and goods [62]. Therefore, there is quite a variety of items that are related to tourism, such as traveling using lodging, transport, and airports, seeing and taking items to and from tourist destinations, including souvenirs, food and drink, insurance, hospital, bank, entertainment, moneychanger, games, and others that visitors may need.

The main goal of the SANEL HERMES analysis was to increase the revenue of tourism destinations by identifying various factors in the tourism environment. Several components, such as Admission paying, Memento shopping, Eating and Drinking, and Staying, can increase more revenue of the destinations. On the other hand, some components of the tourism sector, such as sightseeing places, do not usually generate income, but they are important to attract tourists to the destinations. For these reasons, this model is designed to analyze and develop both the most and the least revenue-generating tourism components at the same time. The following SANEL HERMES components represent the essential requirements for successful tourism:

Sightseeing—traveling to a new place and taking a look at all of its attractions and cool features is a memorable experience.
Admission paying—charges, fees, donations, contributions, or anything else paid for attending an event or entrance to special premises and places.
Night touring—a tour of activities and amenities of a destination that covers the all-night period.
Experiencing—a set of activities in which individuals engage on their personal terms, such as delightful and memorable places.
Learning—knowledge or skill acquired while traveling.
Healing—activities and conditions of a destination to make well again and restore health.
Enjoying—receiving pleasure from tourism activities and services.
Rest and Relaxing—recreation, leisure, and vacation trips (travels) that take place away from a typical environment.
Memento shopping—products and services designed to attract foreign buyers.
Eating and Drinking—food and beverages based on the history, culture, and environment of a particular region.
Staying—providing a room or place from which tourists can engage in any activities at a destination.

The SANEL HERMES tourism model is important for the study because it provides a comprehensive framework for understanding an overview of the key issues that influence the tourism destination. In order to apply the SANEL HERMES model, tourism managers must first evaluate each component and identify which ones are the most important for

their destination. They can then develop strategies to enhance each component and create a more comprehensive and attractive destination that meets the needs and preferences of visitors while also contributing to the economic growth of the destination. As part of the model, the economic, social, and environmental sustainability of the destination is considered. Each component of tourism must be managed so that it does not negatively impact the environment, local communities, or the local economy. The SANEL HERMES model can be applied in different ways, depending on the specific needs and characteristics of each destination. For example, tourism managers can use the model to identify the most popular tourism activities in their destination and develop marketing strategies to promote them. They can also use the model to identify the least popular activities and develop new products or services to attract more visitors. The model can help to support sustainable tourism by identifying components that contribute to the economic, social, and environmental sustainability of the destination, which analysis requires a comprehensive and collaborative approach that involves the participation of tourism stakeholders, including local communities, businesses, governments, and visitors. By providing high-quality and relevant services for each component, visitors are more likely to enjoy their stay and return to the destination in the future.

### 3.1.3. Global RPM/SANEL HERMES Integration Model

The integration model, which combines Global RPM analysis and SANEL HERMES framework of tourism, is an appropriate method for situations of strategic planning. As shown in Figure 1, the synergistic relationship between Global RPM and SANEL HERMES offers the potential for analyzing the development of the tourism sector in a complex and multidimensional environment. Furthermore, the integration method is used within this study to divide the factors that play a vital role in the successful development of sustainable tourism into four groups that are globalization, rationality, professionalism, and morality based on Global RPM analysis. These groups are then further clustered and analyzed into 11 categories—Sightseeing, Admission paying, Night touring, Experiencing, Learning, Healing, Enjoying, Rest and Relaxing, Memento shopping, Eating and Drinking, and Staying according to the system through a SANEL HERMES model. In addition, the tools used in the analyses can be complementary to each other, contributing to a deeper understanding of the environment as a whole. By combining both approaches, it can be better understood how the SANEL HERMES dimensions will increase globally, rationally, professionally, and morally its opportunities and intensities.

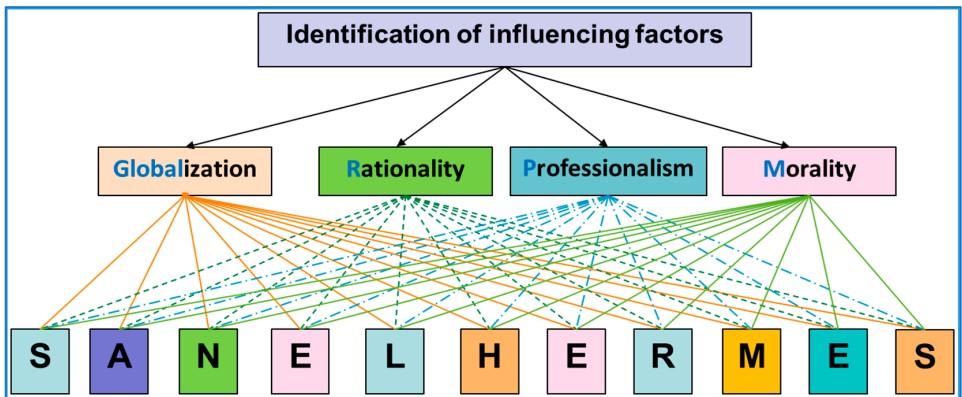

**Figure 1.** Identification of the factors that exert an impact on sustainable tourism development using Global RPM/SANEL HERMES hierarchical integration model. Source: Constructed by the authors.

### 3.2. QSPM Analysis

"Quantitative strategic planning matrix (QSPM) is an analytical tool used to evaluate the relative attractiveness of various strategies based on the key internal and external

factors" [63]. Objectively, QSPM analysis determines and selects more effective strategies than other management methods with less computation [64].

Mainly there are five main steps in the QSPM analysis [63]: (i) it is first necessary to identify the factors that are relevant to the business; (ii) the weights are assigned according to their relative importance. Essentially, all 11 components of Global RPM dimensions respectively must be weighed as a sum to 1.0; (iii) culturalization strategies are chosen and discussed based on appropriately matched factors; (iv) each prospective strategy is assigned an attractive score between 1 and 5 based on an individual examination of every factor, where 1 stands for not attractive, 2 for less attractive, 3 for attractive, 4 for reasonably attractive, and 5 for highly attractive. The scores reflect the current situation of the tourism industry and its challenges regarding sustainability; (v) finally, to calculate the total attractive score, the cumulative value of the scores for each dimension of Global RPM is taken into account, which results in the most effective strategy in terms of practical implementation of the tourism industry.

Research Design, Data Collection, and Analysis

Our study consists of three stages, each of which is approached differently, including interviews with participants and experts and an online tourist survey. This study was conducted using a methodological framework shown in Figure 2.

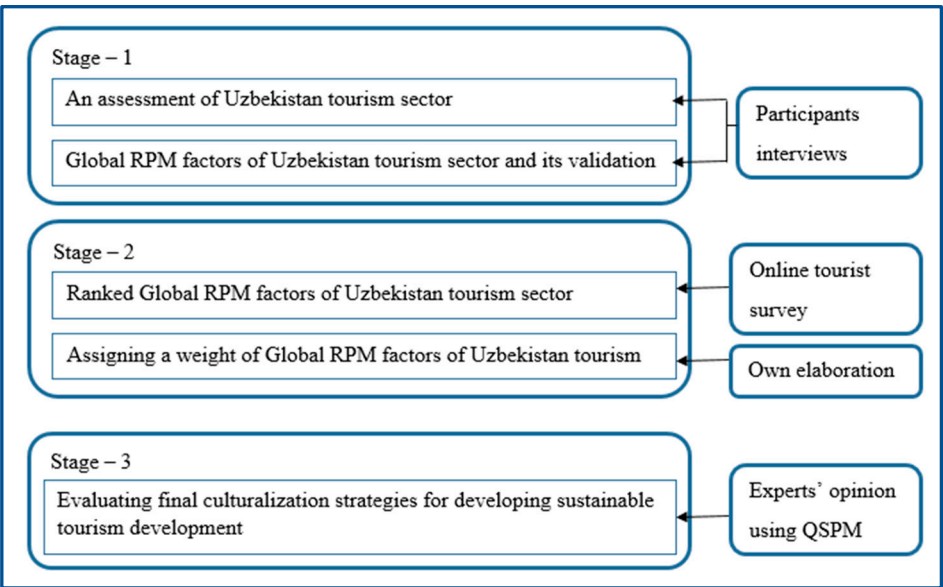

**Figure 2.** Methodological framework. Source: Constructed by the authors.

In the first stage, it was decided that participant interview was used as the primary method of collecting data since the interview method is considered one of the most effective tools to collect data on research topics due to its ability to give participants a voice [65] in where a non-manipulative, extensive, and in-depth description of the research questions is necessary [66]. Participant interviews are important for basic data about the research subject because they provide researchers with an opportunity to gather detailed and comprehensive information directly from individuals who have experienced or are experiencing the phenomenon being studied [67]. Through participant interviews, researchers can ask open-ended questions that allow participants to provide rich and detailed information about their experiences and perspectives. This information can help researchers to identify the key factors that are relevant to the research subject to gain a complete understanding of the phenomenon being studied, such as individual characteristics, social and cultural factors, environmental factors, and other contextual factors [68]. By asking follow-up

questions and probing for more information, researchers can identify gaps in knowledge and develop hypotheses about the factors that may be influencing the research subject [69].

By implementing a purposeful sampling method [70], the authors specifically chose a sample of participants who are well-linked with the tourism sector of Uzbekistan and have adequate knowledge about its operational and functional aspects. There have been 31 in-depth interviews conducted in Uzbekistan, including 2 policymakers, 4 scientists, 5 hotel managers, and 20 entrepreneurs and locals. After 31 interviews, no additional new insights were observed as the saturation point for the data had been reached [71]. During July and November of 2021, the study was conducted by the interviewers as a study exploring the influencing factors (see Table 1) and appropriate strategies (see Table 2) to develop sustainable tourism in Uzbekistan.

**Table 1.** Influencing Factors of Global RPM and SANEL HERMES Dimensions.

| | **Globalization** | **Rationality** | **Professionalism** | **Morality** |
|---|---|---|---|---|
| **S**ightseeing | Ease of travel | Natural beauty and cultural atmosphere | Transport infrastructure conveniences | Conservation of natural ecosystem and cultural heritages |
| **A**dmission paying | International information centers | Price management | Fast service and availability of online purchasing | Politeness and integrity of ticket retailers |
| **N**ight touring | Safety and security | Nightscape beauty and attractions | Urban night amenities | The presence of legal barriers to night travel |
| **E**xperiencing | Research opportunity for arts and cultures | Local activities (e.g., Crafts, traditions, local food production) | Staffs' knowledge of foreign languages and information exchange | Hospitality of the local people |
| **L**earning | International competitiveness in learning | Exclusive style and innovation of learning | Getting relevant degrees in learning | Anticorruption policy |
| **H**ealing | Sustainability and favorable climate | Fauna and forest areas | Opportunity for spiritual development | Environmental safeguarding and green practices of the government |
| **E**njoying | Festivals and events | Innovation and creativity of sustainable entertainment activities | Professionality in organizing sustainable enjoyment activities and services | The prevention of crime |
| **R**est and Relaxing | Diversity of sustainable relaxation amenities | Effectiveness and accessibility of sustainable relaxation activities | Seasonality of relaxation activities and services | Political stability |
| **M**emento and Shopping | International competitiveness of sustainable products and services | Wide cultural base of a memento | Durability and design of sustainable products | Environmental impact of making products |
| **E**ating and Drinking | Gastronomic experience | Tasty and healthy food and drinks | Qualified staff and negotiate effectively | Preventing food waste |
| **S**taying | Convenient location of hotels | Comfort and cleanliness of accommodation | Communication culture and quality service of staff | Environmentally friendly accommodation |

Source: Constructed by the research participants.

**Table 2.** Culturalization strategies combining Global RPM for sustainable tourism of Uzbekistan.

| **Globalization** |
| --- |
| CSG1—Improving and protecting ethnic and cultural diversity, arts and crafts, historic landmark restorations, and incorporating cultural heritage into UNESCO World Heritage Sites |
| CSG2—Promoting export of cultural products, including Uzbek music, films, and books, to familiarize other countries with Uzbek culture |
| CSG3—Enhancing the political, social, and cultural environment for foreign investors in Uzbekistan's tourism market, as well as modifying visa laws and regulations |
| Rationality |
| CSR1—Diversifying sustainable tourism types and strengthening interconnections between tourism and other Regional industries |
| CSR2—Organizing festivals across the country all year round and expanding celebrations and rituals |
| CSR3—Diversifying and improving competitive sustainable tourism services and products |
| Professionalism |
| CSP1—Investing in sustainable road and railway transport infrastructure, healthcare services, hotels and tourist accommodations, and digital technologies |
| CSP2—Hosting international sporting events and developing travel agencies, websites, and apps in different languages to introduce and promote Uzbekistan's tourism destinations |
| CSP3—Establishing roadside and local restaurants, as well as accessible restrooms and tourist information centers for silk road cities to increase sustainable tourism |
| Rationality |
| CSM1—Enhancing local communities and minimizing tourism's negative effects on the environment and society |
| CSM2—Supporting different cultures, ethnicities, races, dialects, and languages while preserving the environment with sustainable principles |
| CSM3—Providing ecotourism training programs with present tourism activities and enhancing tourism assets according to protection-use guidelines |

From the beginning of the data collection process to the end, the interviews were created, refined, and reviewed to provide all of the potential information that contributes to generating useful knowledge. The interview for scientists, policymakers, hotel managers, entrepreneurs, and locals included identification and discussion of the key factors and appropriate culturalization strategies for the tourism sector in Uzbekistan according to previous literature and benchmarking cases which have got successful results. Consequently, the study identified 11 dimensions of SANEL HERMES of destination attributes based on participants' interviews and previous studies. Each dimension consists of four factors in the appropriate case of Global RPM and a total of 44 factors (see Table 1).

After evaluating and selecting each factor, and understanding the association between them, using the Global RPM analysis—four types of strategies, i.e., CSG (Culturalization Strategies for Globalization), CSR (Culturalization Strategies for Rationality), CSP (Culturalization Strategies for Professionalism), and CSM (Culturalization Strategies for Morality), were developed. In particular, Table 2 presents a list of culturalization strategies that aim to promote sustainable tourism in Uzbekistan. These strategies are grouped into four categories that provide guidance as to how to enhance the existing environment in terms of globalization, rationality, professionalism, and morality perspectives. All strategies are offered for every factor instead of concentrating on offering only a type of strategy (CSG, CSR, CSP, and CSM) that corresponds to a dimension (G, R, P, and M, respectively) of the Global RPM. Finally, 12 strategies were identified from benchmarking cases and previous literature for assessing and comparing performance toward the achievement of sustainable tourism development in Uzbekistan. The policymakers may use them to optimize the strategies and identify elements that the tourism sector can implement in the structure.

The second stage consists of two parts. In part one, the results of a tourist survey are presented on a Likert-type scale of 10 factors, while in part two, the factors are weighted according to their ratings on the survey.

During the first part of the second stage, a tourist survey was conducted using questionnaires techniques using online surveys to collect data regarding tourism-based activities in the tourism industry. The research population is domestic and international tourists of Uzbekistan. This study employs a convenient sampling method to collect data. An overall measurement of tourist satisfaction in Uzbekistan is also performed based on a mean of 44 factors. On the 10-point Likert-type scale, each of the 44 factors is rated by online tourist participation. In order to conduct the study, the survey was developed in English and translated into Uzbek. During this online survey, 228 tourists participated. There were 160 males (70% of the total sample) and 68 females (30%). The number of domestic tourists was 195 (85%), while the number of international tourists was 33 (30%). After completing the online tourist survey, prioritization of the influencing factors is conducted according to their achieved rate on the 10-point Likert-type scale. While a sample size of 228 tourists may not be sufficient to provide highly precise estimates of the attitudes and behaviors of the larger population of tourists in Uzbekistan, the survey results can still provide valuable insights into sustainable tourism development in several ways. Firstly, as the survey used a random sampling method, where every tourist in the population has an equal chance of being selected for the survey, then the results of the survey may be more representative of the larger population of tourists. Secondly, since the sample represents the general traveling population in terms of their demographic characteristics, such as age, gender, nationality, and travel preferences, the results of the survey may be more applicable. In summary, the results gained from the survey could still be valuable in providing preliminary information and can inform the design of a larger, more comprehensive survey.

As part of the second stage, part two assigns a weight/rating of influencing factors based on the result of the online tourist survey by their achieved rate using a 10-point Likert-type scale of each factor. Crucially, there must be a sum of 1.0 for the weight of all SANEL HERMES factors related to every dimension of Global RPM. In this study, the opinion of experts and participants is not used to assign weight to the influencing factors. There has been a calculation of the weights of the factors in the study according to the relative importance of the results on the questionnaire using the Likert scale. Fundamentally, the lower a factor obtains a point, the higher it obtains a weight. Because if a factor gets lower points than other factors, it means this factor of the tourism sector needs more attention for prospective strategies to overcome the barrier and meet tourist demands [62]. Indeed, when a parameter of a destination does not match tourist satisfaction, there is a problem with the parameter. Thus, it can be more effective to develop the tourism sector by paying more attention to serious problems [72] and trying to solve them first. Equation (1) shows the calculation of a weight:

$$W_i = \frac{10 - n_i}{(10 - m_j) \times 11} \tag{1}$$

where, W represents weight, *n* is one of 44 mean ratings for the influencing factors according to survey results, while m indicates one of 4 mean ratings for Global RPM dimensions containing 11 factors respectively to n.

i and j stand for the sequence of the mean factors and dimensions correspondingly, while the number 11 is used to form a sum of the weight for every dimension of Global RPM to be equal to 1.0 (as an example, $W_1 + \ldots + W_{11} = 1.0$).

In the third stage, the culturalization strategies are ranked.

Finally, in order to assess the relative attractiveness of culturalization strategies based on the key factors by QSPM using Global RPM dimensions, some study policymakers, scientists, and hotel managers were contacted further. In addition, a brief cover letter is provided to introduce the research topic, explain the purpose of the study, and get agreement for using information. For evaluating attractive scores, the primary reason for contacting only policymakers, hotel managers, and scientists was their experience

and knowledge regarding the study's subject and scientific aspect. Lastly, culturalization strategies were explored and proposed with the goal of developing sustainable tourism in Uzbekistan.

## 4. Results

For the development of sustainable tourism in Uzbekistan, it has been proposed to carry out a research study that focuses on two main objectives; (i) conducting Global RPM analysis and SANEL HERMES model of the tourism industry in Uzbekistan, and (ii) offering culturalization strategies that support sustainable tourism development in the country. Therefore, in-depth interviews were conducted by the authors to obtain the relevant information, and using both Global RPM and QSPM methods, the authors identified, evaluated, and then offered culturalization strategies based on the findings. It is the purpose of this section to present results and findings that have been obtained from analyzing conventionally the interview transcripts as well as the tourist survey responses in the form of the influencing factors and appropriate strategies.

### 4.1. Global RPM and SANEL HERMES Analysis of Tourism Sector in Uzbekistan

In the case of Uzbekistan's tourism sector, the factors of Global RPM have been identified. Moreover, Global RPM's performance is measured by four factors from each SANEL HERMES dimension. Table 1 provides an overview of all influencing factors used in the research. The data analysis was based on an inductive approach, given the exploratory nature of the study. As part of the analysis, 11 factors of SANEL HERMES were pertaining to every Global RPM dimension respectively (globalization, rationality, professionalism, and morality) (see Table 3) with the satisfaction of tourists by giving scores from a minimum of one point (very unsatisfied) to maximum 10 points (very satisfied). Finally, the total average point of tourist satisfaction stands at 6.49 out of 10 for all factors. In terms of factors affecting dimensions of Global RPM, rationality scored the highest with 6.95, while professionalism scored the lowest with 6.06. It is evident from the results that tourism activities still face a number of obstacles in order to achieve sustainability, and there is a need for a better governance system based on professionalism and morality.

**Table 3.** Descriptive results of Global RPM and SANEL HERMES Dimensions.

|  | Globalization | Rationality | Professionalism | Morality | Mean: |
|---|---|---|---|---|---|
| Sightseeing | 7.18 | 8.35 | 5.65 | 8.46 | 7.41 |
| Admission paying | 5.64 | 6.29 | 5.12 | 5.96 | 5.75 |
| Night touring | 7.75 | 7.41 | 6.55 | 6.40 | 7.03 |
| Experiencing | 6.61 | 7.84 | 6.10 | 6.36 | 6.73 |
| Learning | 5.36 | 5.25 | 5.36 | 4.44 | 5.10 |
| Healing | 6.83 | 6.68 | 6.23 | 5.29 | 6.26 |
| Enjoying | 6.63 | 5.89 | 6.33 | 6.71 | 6.39 |
| Rest and Relaxing | 6.42 | 6.49 | 6.13 | 6.26 | 6.33 |
| Memento shopping | 5.88 | 7.05 | 6.18 | 6.15 | 6.32 |
| Eating and Drinking | 7.62 | 8.24 | 6.43 | 6.74 | 7.26 |
| Staying | 6.74 | 6.96 | 6.62 | 7.14 | 6.87 |
| Mean: | 6.60 | 6.95 | 6.06 | 6.36 | 6.49 |

Note: All factors were rated on a 10-point scale, where 1 and 2 stand for very unsatisfied, 3 and 4 for moderately dissatisfied, 5 and 6 for neutral, 7 and 8 for moderately satisfied, and 9 and 10 for very satisfied.

For globalization, the results of Global RPM (see Table 3) indicate that the most essential factors that received the highest point were "Safety and security", followed by

"Gastronomic experience" and "Ease of travel". In contrast, "International competitiveness in learning" had the lowest weight.

For rationality, "Natural beauty and cultural atmosphere" and "Tasty and healthy food and drinks" have obtained the highest points, followed by "Local activities (e.g., Crafts, traditions, local food production)" and "Nightscape beauty and attractions". On the contrary, "Exclusive style and innovation of learning" and "Innovation and creativity of sustainable entertainment activities" had the lowest points. Based on the factors discussed above, rationality factors achieved the highest score, which means only the rationality of the tourism sector moderately satisfied the tourists. It can be seen that visiting Uzbekistan was moderately rational according to tourist survey results.

Further, for professionalism and morality, the opinion of tourists remained neutral and did not meet even moderate satisfaction with the lowest point. For professionalism, "Urban night amenities" had the highest points, followed by "Qualified staff and negotiate effectively in eating and drinking" and "Professionality on organizing sustainable enjoyment activities and services". In contrast, "Fast service and availability of online purchasing" and "Professionality on organizing sustainable enjoyment activities and services" had the lowest weights.

For morality, "Conservation of natural ecosystem and cultural heritages" obtained the highest point, followed by "Environmentally friendly accommodation" and "Preventing food waste". On the contrary, the "Anticorruption Policy" was ranked lowest. During the assessment of morality factors, the overall average point was 6.36 out of 10. It suggests that there is a need to pay more attention to the difficulties associated with the moral aspects of the tourism sector.

In the study, the dimensions of Global RPM were also correlated to analyze the relationship between them. Correlation analysis is a very useful tool for exploring our data and quickly obtaining an overview of what the relationships look like. A correlation matrix is often a good feature of research at the beginning of the results section, as it makes it easier for the reader to understand what is going on in the more advanced analyses. According to Table 4, all dimensions are highly correlated. For example, the correlation between globalization and professionalism had the highest coefficient with 0.7167, while the correlation between globalization and morality obtained the lowest one with 0.637. Generally, the correlations between dimensions are close to each other with a slight difference of approximately 0.08 coefficient.

**Table 4.** Correlation matrix of independent variables.

|  | Globalization | Rationality | Professionalism | Morality |
|---|---|---|---|---|
| Globalization | 1.0000 |  |  |  |
| Rationality | 0.7120 | 1.0000 |  |  |
| Professionalism | 0.7167 | 0.6609 | 1.0000 |  |
| Morality | 0.6371 | 0.6416 | 0.6490 | 1.0000 |

*4.2. The Assignment of Weights for the Global RPM Factors*

Based on the questionnaire results, the factors have been weighted in order of importance. The assignment of weights aims efficiently to determine every culturalization strategy based on factors examined individually concerning the current challenges of sustainability facing the tourism industry. Specifically, a factor with a lower point value is given a higher weight and vice versa for a higher point. The reason is that if a tourism factor receives a lower satisfaction score than other factors, it can be seen as a serious barrier and receive more focus. Table 5 shows the assignment of weights for Global RPM and SANEL HERMES Dimensions. As a consequence of the Global RPM analysis carried out, a total of 44 factors were determined, with 11 for each globalization, rationality, professionalism, and morality group. In contrast to survey results, the factors that obtained lower points

had higher points with weights between 0.04 and 0.14 coefficient. In terms of the learning dimensions for SANEL HERMES, the four factors had the lowest points among respondents, but the factors generated the highest coefficients after considering their weights with 0.12, 0.14, 0.11, and 0.14 for globalization, rationality, professionalism, and morality, respectively. In the factors of sightseeing, eating and drinking dimensions had the lowest weights, ranging from 0.04 to 0.08.

**Table 5.** The assignment of weights for Global RPM and SANEL HERMES Dimensions.

|  | Globalization | Rationality | Professionalism | Morality |
|---|---|---|---|---|
| S | 0.08 | 0.05 | 0.10 | 0.04 |
| A | 0.12 | 0.11 | 0.11 | 0.10 |
| N | 0.06 | 0.08 | 0.08 | 0.09 |
| E | 0.09 | 0.06 | 0.09 | 0.09 |
| L | 0.12 | 0.14 | 0.11 | 0.14 |
| H | 0.08 | 0.10 | 0.09 | 0.12 |
| E | 0.09 | 0.12 | 0.08 | 0.08 |
| R | 0.10 | 0.10 | 0.09 | 0.09 |
| M | 0.11 | 0.09 | 0.09 | 0.10 |
| E | 0.06 | 0.05 | 0.08 | 0.08 |
| S | 0.09 | 0.09 | 0.08 | 0.07 |
| Total: | 1.0 | 1.0 | 1.0 | 1.0 |

Note: The factors are given as coefficients between 0 and 1 according to survey results; it stands for how much it will be important to measure the actual barriers. This coefficient represents the significance of the factor.

*4.3. QSPM Analysis*

Further analyses of QSPM were conducted with the goal of ranking the strategies on the basis of their overall attractiveness (Table 6). According to the results of the QSPM analysis, the most effective strategy for sustainable tourism in Uzbekistan is CSP1, i.e., "Investing in sustainable road and railway transport infrastructure, healthcare services, hotels and tourist accommodations, and digital technologies". Consequently, the CSM2, CSR1, CSR2, CSG3, CM1, and CSR3 strategies with the highest total attractiveness scores accordingly can transform the incongruous practices in sustainable tourism development. Despite the fact that all the culturalization strategies suggested in this study are feasible to implement, considering the final total attractive score, it is possible to determine which of the strategies and in which priority they may be most practical, which the associated stakeholders and policymakers should consider in order to ensure the sustainability of future tourism development.

**Table 6.** Final Ranking of Strategies using QSPM method.

|  | Globalization | Rationality | Professionalism | Rationality | Mean | Rank |
|---|---|---|---|---|---|---|
| CSG1 | 2.44 | 2.10 | 1.58 | 1.76 | 1.97 | 11 |
| CSG2 | 2.43 | 2.17 | 2.34 | 1.37 | 2.08 | 10 |
| CSG3 | 3.48 | 2.88 | 2.98 | 3.06 | 3.10 | 5 |
| CSR1 | 2.98 | 3.49 | 3.04 | 3.36 | 3.22 | 3 |
| CSR2 | 3.01 | 3.06 | 3.67 | 2.72 | 3.12 | 4 |
| CSR3 | 2.76 | 3.43 | 2.71 | 1.82 | 2.68 | 7 |

**Table 6.** *Cont.*

|  | **Globalization** | **Rationality** | **Professionalism** | **Rationality** | **Mean** | **Rank** |
|---|---|---|---|---|---|---|
| CSP1 | 3.21 | 3.27 | 3.98 | 2.88 | 3.34 | 1 |
| CSP2 | 2.59 | 1.80 | 3.47 | 1.95 | 2.45 | 9 |
| CSP3 | 2.62 | 2.32 | 3.06 | 1.94 | 2.48 | 8 |
| CSM1 | 2.74 | 2.61 | 2.73 | 3.30 | 2.84 | 6 |
| CSM2 | 3.20 | 3.23 | 2.97 | 3.83 | 3.31 | 2 |
| CSM3 | 1.57 | 1.46 | 1.77 | 1.52 | 1.58 | 12 |

Note: All strategies is assigned an attractive score between 1 and 5, where 1 stand for not attractive, 2 for less attractive, 3 for attractive, 4 for reasonably attractive, and 5 for highly attractive.

## 5. Discussion

It is widely accepted that Uzbekistan's tourism sector and its potential are seen as one of the driving forces for the country's economic development [60]. However, taking note of the study's findings that there are several factors that make it unsustainable for its tourism sector, stability, and ecology. In fact, there has been an increase in tourists who are interested in activities associated with conservation and nature in recent years. Therefore, it is imperative to link Uzbekistan's tourism potential with future development planning and sustainable perspectives in order to maximize the potential of tourism in the region by increasing the volume of tourism activities and investing in tourism infrastructure.

On the one hand, some factors of the tourism sector of Uzbekistan achieved high satisfaction of tourists such as the conservation of the natural ecosystem and cultural heritages, natural beauty, and cultural atmosphere, tasty and healthy food and drinks, and Local activities (e.g., Crafts, traditions, local food production). In fact, Uzbekistan's tourism promotion relies heavily on its historical and cultural sites and major cities, such as those along the ancient Silk Road [73]. Moreover, the most significant advantages of the tourism sector can be seen as natural and cultural sightseeing places and the experience of the culinary arts. Moreover, one way in which sustainability can be achieved is through gastronomic tourism, which contributes to the development of tourism in a sustainable manner [74]. Due to this truth, it is one of the most popular practices in tourism these days, and it promotes gastronomic tourism. Consequently, these advantages suggest that there are valuable opportunities for the development of Uzbekistan's tourism sector in a sustainable way and provide prospects for integrating tourism with sustainable planning.

However, in order to carry out a comprehensive analysis of the tourism industry in Uzbekistan and suggest culturalization strategies with a view to tying the potential for tourism in the destination to future strategic development and sustainable practices, the results of the Global RPM analysis reveal that the factor of anticorruption policy is currently representing most pressing issue in the development of sustainable tourism in Uzbekistan. As a result, this challenge is becoming more complex for sustainability and tourism development. Corruption in several industries of Uzbekistan is a vital issue that has long-term damaging effects on society [75]. Therefore, the reduction of corruption will significantly increase tourism competitiveness, as clear from previous literature [76,77]. The next most pressing issue, which obtained the second least point, was fast service and availability of online purchasing has seen a lack of tourist services and digitalization of the sector. Having access to digital technology throughout the planning process and the duration of the traveling itself has made traveling easier and more affordable than ever [78]. Consequently, digital technology can offer visitors real-time information on travel and fast services and provide personal reviews [72,79].

Furthermore, the findings of the study indicate that the factors of sustainable relaxation and enjoyment activities, durability and design of sustainable products, exclusive style and innovation of sustainable learning, environmental safeguarding and green practices of the government, and making products do not support the sustainability of the tourism sector in

the region. For the reasons that Uzbekistan has suffered severe pollution due to industrial waste, pesticides in agriculture, fertilizer, and the Aral Sea catastrophe [80–82]. There is a need for regional sustainable development to ensure environmental and ecological security. Furthermore, it is not only important that the government considers it a serious problem, but the local residents must also take the lead in spearheading sustainable initiatives, especially with regard to conserving and protecting the region's environmental resources. In general, the results of the study reveal that Uzbekistan's tourism sector failed to meet even moderate sustainability standards as measured by tourist satisfaction. Thus, this destination is facing serious issues in developing sustainable tourism in a holistic way. Nevertheless, the government and stakeholders in Uzbekistan may be able to make use of the findings of Global RPM and QSPM analysis in more practical ways for sustainable tourism development.

Accordingly, it is claimed in the study that in order to maintain Uzbekistan's stability and grow its tourism industry, the current practices, management, and existing activities of tourism cannot be viewed as sustainable. In this regard, it is possible to transform the existing incongruous practices and activities of Uzbekistan tourism into a strong basis for sustainable development if the policymakers act in accordance with the identified strategies. In particular, the CSP1 and CSM2 strategies, with the highest rank that represents professionalism and morality aspects, provide a practical framework to develop and expand Uzbekistan's tourism industry through sustainable tourism strategies as well as promoting environmentally friendly and empathetic ethics among visitors. Moreover, The CSG strategies aimed at developing globalization of the tourism sector, are vital to the CSG3 strategy, which is improving political, social, and cultural conditions for the presence of foreign investors in the tourism market of Uzbekistan and modifying visa laws and regulations. Further, CSR strategies, a tool to improve rationality, are practical in diversifying sustainable tourism types, strengthening interconnections between tourism and other regional industries and reviving the celebrations and rituals, and holding festivals all year round all over the country. Finally, Uzbekistan can use culturalization strategies to develop policy frameworks and defensive plans to achieve long-term sustainability and economic growth.

By applying the Global RPM analysis and SANEL HERMES model, as well as the QSPM based on the main objectives of sustainable tourism and identifying the vision statement for the country, this study presented a methodological approach for managing the sustainable development of tourism destinations and its processes. In fact, for sustainable tourism to be achieved, it is essential for effective strategies to be developed and implemented as a result of collaboration between stakeholders, policymakers, and organizations that are relevant to the industry. In addition to the vision statement, sub-criteria and priority criteria must be considered when developing tourism strategies. To achieve the maximum benefit from sustainable tourism, culturalization strategies should be assessed from a holistic perspective.

## 6. Conclusions

With an aim to develop sustainable tourism in Uzbekistan, the study—using Global RPM analysis integrated with the SANEL HERMES model and QSPM—analyzed the tourism sector and suggested prospective strategies. As a result of the study, it has been found that Uzbekistan's tourism sector is continuously expanding and progressing. However, anticorruption policy, fast service, availability of online purchasing, transport infrastructure, the international competitiveness of sustainable products and services, environmental safeguarding and green practices of the government, and innovation and creativity of sustainable entertainment activities are not sustainable for the tourism industry, stability, and ecology of Uzbekistan. As a result, it can be concluded that if the policymakers follow the proposed strategies, as in the case of Uzbekistan tourism in the region, the current tourism incongruities can be transformed into opportunities for sustainable development. In this regard, the following policy recommendations are made:

1. Officials should encourage sustainable tourism development on the identified comparative advantages and challenges and sustainable management of natural and cultural resources;
2. Government and tourism authorities should encourage innovation and investment in sustainable tourism by providing the political, social, and cultural environment for businesses that develop sustainable tourism products and services;
3. Collaboration and partnerships between governments, tourism authorities, businesses, and local communities should be fostered through platforms that enable stakeholders to exchange ideas and share best practices;
4. Investing in sustainable infrastructure, quality roads and vehicles, hotels and tourist accommodations, and digital technologies in tourist areas should be focused on maximizing the potential of foreign investors, as well as strengthening interconnections between tourism and other regional industries;
5. Modern advertising and marketing technologies should be utilized to boost tourism by diversifying and improving competitive, sustainable tourism services and products.

In this study, several key contributions are made to the literature on sustainable tourism. Firstly, this study contributes to the literature on Uzbekistan's tourism industry and sustainable tourism development. Particularly, it is imperative that the true picture of Uzbekistan is painted to tourists as it forms the basis for them to consider Uzbekistan as a sustainable tourism destination. However, the literature on sustainable tourism development in Uzbekistan appears to be lacking. Using the framework presented in this study, it is possible to develop Uzbekistan's tourism sector on the basis of sustainable tourism. Accordingly, the findings of this paper may serve as a framework for further research to explore sustainable tourism in Uzbekistan. Secondly, it attempts to link Uzbekistan tourism's key factors to culturalization strategies that can contribute to sustainable future development prospects. As well as several studies have been conducted on development strategies related to culture [83,84]. Nevertheless, many studies have not specified culturalization strategies, especially when it comes to sustainable tourism. This paper defines the term culturalization strategies differently and discusses the previous literature. Consequently, the strategies can assist in the implementation the sustainable development, which has been formulated in accordance with the cultures. Additionally, the study contributes scientifically to the tourism literature in order to add insights into the theories and models developed for culturalization strategies of sustainable tourism. Finally, innovative contributions of this study's methodology include presenting a comprehensive view that can assist with the design and integration of strategic decision-making procedures and improving the quantitative and qualitative side of strategic planning using the pioneering application of Global RPM analysis in combination with SANEL HERMES tourism model and the QSPM analysis. They may also contribute to the theoretical and conceptual development of the field by proposing new frameworks, models, or theories for understanding the complex relationships between culture, sustainability, and tourism development. Furthermore, this study contributes to understanding Uzbekistan's tourism potential in relation to sustainable prospects through exploring and proposing strategies. It must be said that previous tourism literature has not adequately addressed specific sustainable factors that motivate tourists, resulting in a knowledge gap. In this regard, this paper attempts to fill the gap between tourism prospects and sustainability concerning tourists' perceptions and essentially preserving cultural, social, natural, and historical assets, which are a necessity to create sustainable tourism in the destination.

The study has been conducted with the objective of focusing on the entire country of Uzbekistan and its tourism sector. Thus, it is necessary to acknowledge a number of limitations. Regarding this, one limitation is that it is only focused on Uzbekistan and may not be directly applicable to other countries. The specific cultural, social, economic, and environmental contexts of Uzbekistan may differ from those of other countries, and therefore the results of this study may not be relevant to other locations. However, the study's methodology and framework for analyzing effective strategies for promoting

sustainable tourism, including the use of Global RPM and QSPM analysis, may be useful for similar studies in other countries. Therefore, while the specific results of this study may not be generalizable to other contexts, the overall approach can be applied and modified to suit the needs of other countries. This limitation should be taken into consideration when interpreting and applying the findings of this study to other contexts. Furthermore, It has been decided that the selection of a sampling framework and approach was meant to provide greater insight into how the current inappropriate activities of the tourism sector could be transformed into sustainable activities instead of providing conclusions that can be individually applied to each city's development strategies of Uzbekistan. Based on the results of this comprehensive study, it appears that Uzbekistan tourism's inappropriate activities can be transformed into sustainable development prospects by using the identified strategies. Additionally, this study also identifies several research avenues that could be further explored in the future. Firstly, to make a better understanding of the potential of tourism in Uzbekistan and to gain a better insight into the overall development of the sector, future research studies should be conducted on the tourism challenges and issues for each main touristic city of the country. Secondly, for the promotion of sustainable production and consumption, it is necessary to analyze in detail the environmental impact of different sectors of the economy, which are closely related. Namely, sustainable strategies require a deep knowledge of how economic activities, such as extraction, agriculture, manufacturing, trade, and services, interact with each other. Moreover, future research could also be based on a sector-driven approach of Global RPM analysis to achieve sustainable development by identifying the direct and indirect impacts of other sectors on tourism.

**Author Contributions:** Conceptualization, J.Y.J.; methodology, J.Y.J.; software, M.K.; validation, Y.S.; formal analysis, O.S.; investigation, M.K.; resources, P.M.; data curation, Y.S. and P.M.; writing—original draft preparation, M.K.; writing—review and editing, O.S.; visualization, Y.S.; supervision, M.K.; project administration, O.S.; funding acquisition, P.M. All authors have read and agreed to the published version of the manuscript.

**Funding:** The APC was funded by ZHAW.

**Institutional Review Board Statement:** Not applicable.

**Informed Consent Statement:** Not applicable.

**Data Availability Statement:** Not applicable.

**Conflicts of Interest:** The authors declare no conflict of interest.

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
