# Peer review of "Evaluating Culturalization Strategies for Sustainable Tourism Development in Uzbekistan"

_sustainability, doi:10.3390/su15097727_

Round 1

Reviewer 1 Report

First of all, congratulations for your work. However, manuscript should be modified following considerations described as follows:

Line 2-3. Please do not use acronyms in title.

Lines 17-34. Please do not use acronyms in abstract either.

Line 96. … our primary objective is… Then, what is your secondary objective? It is not clear. Please state sentence as in abstract: this research aims to…

Line 98. You must state complete version of acronym first time it appears. Please deeper explain what SANEL HERMES is.

Line 126. Avoid pronouns (He/She, etc.), please remain impersonal/use of passive instead.

Line 137. ANP mistyping for AN.

Line 219-244. I cannot understand Figure 1. What SANEL HERMES means?

Line 265. You must explain what SANEL HERMES means before to avoid confuse readers. Please briefly state all acronyms used in abstract and introduction.

Line 305. You must number formulas.

Line 314. Please displace Table 2 under the paragraph you are referring to. Please also deeper explain Table 2.

Line 515. Avoid “we” and/or “our” words, please remain text impersonal.

Line 525. Please be consistent in table design. Avoid other text colour than black.

Line 706. Please state as limitation that this study is only Uzbekistan-centred and how results can be useful in similar countries. Also, a low number of surveys has been used. Please demonstrate how this number can be enough to get representative results (e.g., Stating total number of tourists in Uzbekistan per year, etc.)

Line 733. References cannot follow MDPI style requirements please revise.

Author Response

Dear Editor,

Our sincere gratitude goes out to you for giving us this opportunity to submit a revised draft of the manuscript “Evaluating Culturalization Strategies for Sustainable Tourism Development in Uzbekistan” for publication in the Sustainability. We appreciate you and the reviewers for your precious time in reviewing our paper and providing valuable and insightful comments, which contributed to the current version's improvement. The authors have carefully considered the comments and tried our best to address every one of them. We hope the manuscript after careful revisions meet your high standards. The authors welcome further constructive comments if any.

Please see below for a point-by-point response to the reviewers’ comments and concerns.

Sincerely,

The Authors

Reviewer 2 Report

The article "Culturalization Strategies for Sustainable Tourism Development in Uzbekistan: An Application of Global RPM Analysis and SANEL HERMES Model" by Ji Young Jeong et al, presents a thorough analysis of the culturalization strategies for sustainable tourism development in Uzbekistan.

The introduction is well written and provides a clear overview of the importance of sustainable tourism development in the current economic and policy landscape. The methods are clearly stated, and the use of a hybrid method of Global RPM and SANEL HERMES is effectively employed to identify and classify influencing factors that contribute to the existing barriers to sustainable tourism. The results and discussion sections provide an insightful analysis of the data gathered through the use of questionnaires and expert evaluations, and the proposed culturalization strategies are relevant and well-evaluated using a Quantitative Strategic Planning Matrix.

Overall, I think that this article would be of great interest to the readers of Sustainability. The study provides a valuable contribution to the field of sustainable tourism development in Uzbekistan, highlighting the need for policymakers to consider the proposed culturalization strategies to overcome the barriers to sustainable tourism. The article is well-written, and the results and discussion sections are effective in conveying the significance of the study's findings.

Therefore, I recommend that the article be accepted for publication in Sustainability.

Author Response

(The authors gave the same response as above.)

Reviewer 3 Report

The paper is well written and it was very interesting to read the paper. The paper's topic and conducted research are very important and justified to be presented in a high-quality Journal. The subject is very important for the literature. However, some issues need to be addressed carefully. Please consider the following comments to enhance the quality of the paper.

*The authors should point out the research gap for this study in the Introduction.

* The authors did not include a detailed rationale for the choice of the subject of the study, including basic data about the research subject.

*Please explain in the paper the extent to which the developed paper can be relevant to the international scientific field.

* The study lacks an indication of the audience of the data? Who could use the presented study (actors, recipients)? To whom can this knowledge be practical. 

*In conclusion, please indicate to what extent the proposed paper is innovative and what is its scientific contribution in the field of Sustainable Tourism ? 

* The study should indicate recommendations in a time horizon. What do the authors think should be the priority in realizing Sustainable Tourism?

Author Response

(The authors gave the same response as above.)

Reviewer 4 Report

Thank you for allowing me to review this article. My proposals to improve the paper are:

- the title should be short and informative. Therefore, shorten the title

- do not use abbreviations in the abstract

- spell all the abbreviations when arising for the first time, like lines 98,99, 206-207

- something is missing in the sentence in lines 48-49

- research question and hypotheses are missing in the intro section

- write more about the methodology Global RPM and SANEL HERMES

- I see this article as very confusing. It would be beneficial to use only one method to collect the data (survey or interviews). It is hard to read the results section because I did not get the methodology correct and I do not know which data you are analysing- therefore, what is the research gap?

-first interviews, second survey, third new letters (lines 317-324). no, go. This is a science not management in the business

-move section 3.2 to the beginning of the methodology section

- what are the limitations and delimitations of the study

- policy and managerial implications?

- add 5 or more most recent citations (2022, 2023)

- paper is way too long and it needs a clearer focus. For now, the paper is confusing, and the authors are trying to publish the research.

Good luck.

Author Response

(The authors gave the same response as above.)

Reviewer 5 Report

Ji Young Jeong, Mamurbek Karimov, Yuldoshboy Sobirov, Olimjon Saidmamatov and Peter Marty. Culturalization Strategies for Sustainable Tourism Development of Uzbekistan: Global RPM Analysis and SANEL HERMES Model Approach

The article “Culturalization Strategies for Sustainable Tourism Development of Uzbekistan: Global RPM Analysis and SANEL HERMES Model Approach” by Ji Young Jeong, Mamurbek Karimov, Yuldoshboy Sobirov, Olimjon Saidmamatov and Peter Marty aims to identify the most critical issues and barriers to sustainable tourism development in Uzbekistan and proposes the most effective strategies to overcome those barriers. The study results will affect the development of future regional policies that stakeholders must adopt. In addition, they define other destinations that are interested in developing sustainable tourism.

In the abstract part of the article, Culturalization Strategies for Sustainable Tourism Development of Uzbekistan: Global RPM Analysis and SANEL HERMES Model Approach, written main objectives of the research, research methods and data collection methods, main results of the study, novelty of the study, implications they show, how to benefit from the study.

The article was written on Culturalization Strategies for Sustainable Tourism Development in Uzbekistan. In the article, the Author revealed the objective necessity of research at the global and regional levels. The goals and tasks of the research were clearly defined, and the set goals and tasks were achieved.

In the literature review, the authors analyzed and did chronological and theoretical research on Culturalization Strategies for Sustainable Tourism Development of Uzbekistan and experiences in other developed countries.

 Methodology, given Global RPM and SANEL HERMES analysis of the tourism industry in Uzbekistan, provided culturalization strategies designed to enhance sustainable tourism development while being applicable to situations of strategic planning. Moreover, the Author analyzed the Influencing Factors of Global RPM and SANEL HERMES Dimensions.

Results the Author, after profoundly analyzing the object and getting Descriptive results of Global RPM and SANEL HERMES Dimensions and According to the results of the QSPM analysis, found the most effective strategy for sustainable tourism in Uzbekistan.

Overall, the paper can be accepted with minor changes.

1. Please state clearer the research design, questions, hypotheses and methods.

2. 72 citations are not enough. Minimum 80 needed.

E.g. For new technologies:

Novotny, A., David, L., Csafor, H. (2015). Applying RFID technology in the retail industry – benefits and concerns from the consumer’s perspective

Amfiteatru Economic 17(39). pp. 615-631. 17 p.

https://www.amfiteatrueconomic.ro/RevistaDetalii_EN.aspx?Cod=57

https://www.emerald.com/insight/content/doi/10.1108/978-1-83982-688-720201017/full/html

Etc.

Author Response

(The authors gave the same response as above.)

Round 2

Reviewer 1 Report

Line 169. He/She issue has not been solved.

Line 180. ANP mistyping has not been solved.

Line 584. Please do not refer to Equation 1 as “the following equation”. Please state as: Equation 1 shows calculation…

Author Response

Dear reviewer,

Thanks for your comments.

I corrected the mentioned mistakes, highlighted in the manuscript please: 

1) Line 169. He/She issue has not been solved. - DONE

2) Line 180. ANP mistyping has not been solved. - DONE

3) Line 584. Please do not refer to Equation 1 as “the following equation”. Please state as: Equation 1 shows calculation… - DONE

Reviewer 4 Report

Dear Authours.

You have improved the paper. But, some answers from the authors are not correct, for example:

"Thanks for your kind reminders. In response to
the reviewer's suggestion, we have included and
added limitations in the conclusion section [page
30, line 1168-1178]."

You could improve the paper (like adding further research recommendations, which is now stopped in line 150) and credibly answer my previous recommendations. For now, as an example, there is no page 30, and also lines 1168-1178 do not exist. Etc.

Please send me the latest answer to my concerns from the first review.

Author Response

Dear Reviewer,

We are working on TRACK mode, thus there is page 30, line 1168-1178. For your convenience, I highlighted this part in yellow color. This is in paragraph starting with “The study has been conducted ….” (3rd paragraph of conclusion part)

In the conclusion part (1st and 3rd paragraphs), future recommendations are integrated. Highlighted in yellow for your convenience please.

Latest version of the paper is uploaded to the system.